# Growth strategy of microbes on mixed carbon sources

Xin Wang [1,2], Kang Xia [1,3], Xiaojing Yang [1] & Chao Tang [1]

A classic problem in microbiology is that bacteria display two types of growth behavior when cultured on a mixture of two carbon sources: the two sources are sequentially consumed one after another (diauxie) or they are simultaneously consumed (co-utilization). The search for the molecular mechanism of diauxie led to the discovery of the *lac* operon. However, questions remain as why microbes would bother to have different strategies of taking up nutrients. Here we show that diauxie versus co-utilization can be understood from the topological features of the metabolic network. A model of optimal allocation of protein resources quantitatively explains why and how the cell makes the choice. In case of co-utilization, the model predicts the percentage of each carbon source in supplying the amino acid pools, which is quantitatively verified by experiments. Our work solves a long-standing puzzle and provides a quantitative framework for the carbon source utilization of microbes.

[1] Center for Quantitative Biology, School of Physics and Peking-Tsinghua Center for Life Sciences, Peking University, Beijing 100871, China. [2] Channing Division of Network Medicine, Brigham and Women's Hospital and Harvard Medical School, Boston, MA 02115, USA. [3] College of Life Sciences, Wuhan University, Wuhan 430072, China. These authors contributed equally: Xin Wang, Kang Xia. Correspondence and requests for materials should be addressed to C.T. (email: tangc@pku.edu.cn)

During the course of evolution, biological systems have acquired a myriad of strategies to adapt to their environments. A great challenge is to understand the rationale of these strategies on quantitative bases. It has long been discovered that the production of digestive enzymes in a microorganism depends on (adapts to) the composition of the medium[1]. More precisely, in the 1940s Jacques Monod observed two distinct strategies in bacteria (*E. coli* and *B. subtilis*) to take up nutrients. He cultured these bacteria on a mixture of two carbon sources, and found that for certain mixtures the bacteria consume both nutrients simultaneously while for other mixtures they consume the two nutrients one after another[2,3]. The latter case resulted in a growth curve consisted of two consecutive exponentials, for which he termed this phenomenon diauxie. Subsequent studies revealed that the two types of growth behavior, diauxic- and co-utilization of carbon sources are common in microorganisms[4–8]. The regulatory mechanism responsible for diauxie, that is the molecular mechanism for the microbes to express only the enzymes for the preferred carbon source even when multiple sources are present, is commonly ascribed to catabolite repression[5,9–13]. In bacteria it is exemplified by the *lac* operon and the cAMP-CRP system[14–17]. In yeast, the molecular implementation of catabolite repression differs, but the logic and the outcome are similar[5].

Why have microbes evolved to possess the two strategies and what are the determining factors for them to choose one versus the other? For unicellular organisms, long term survival and growth at the population level are paramount. Cells allocate their cellular resources to achieve optimal growth[18–27]. In particular, it has been demonstrated that the principle of optimal protein resource allocation can quantitatively explain a large body of experimental data[20,21] and that the most efficient enzyme allocation in metabolic networks corresponds to elementary flux mode[25,26,28]. In this paper, we extend these approaches to address the question of multiple carbon sources and show that the two growth strategies can be understood from optimal enzyme allocation further constrained by the topological features of the metabolic network.

## Results

### Categorization of carbon sources.
Carbon sources taken by the cell serve as substrates of the metabolic network, in which they are broken down to supply pools of amino acids and other components that make up a cell. Amino acids take up a majority of carbon supply (about 55%)[29–31]. As shown in Fig. 1, different carbon sources enter the metabolic network at different points[31]. Denote those sources entering the upper part of the glycolysis Group A and those joining at other points of the metabolic network Group B (Fig. 1). Studies have shown that when mixing a carbon source of Group A with that of Group B, the bacteria tend to co-utilize both sources and the growth rate is higher than that with each individual source[6,7,32]. When mixing two sources both from Group A, the bacteria usually use a preferred source (of higher growth rate) first[4,6,11,13,33,34].

### Precursor pools of biomass components.
Based on the topology of metabolic network (Fig. 1), we classify the precursors of biomass components (amino acids and others) into seven precursor pools. Specifically, each pool is named depending on its entrance point on the metabolic network (see Supplementary Note 1.3 for details): a1 (entering from G6P/F6P), a2 (entering from GA3P), a3 (entering from 3PG), a4 (entering from PEP), b (entering from pyruvate/Acetyle-CoA), c (entering from α-Ketoglutarate) and d (entering from oxaloacetate). The Pools a1-a4 are collective called as Pool a.

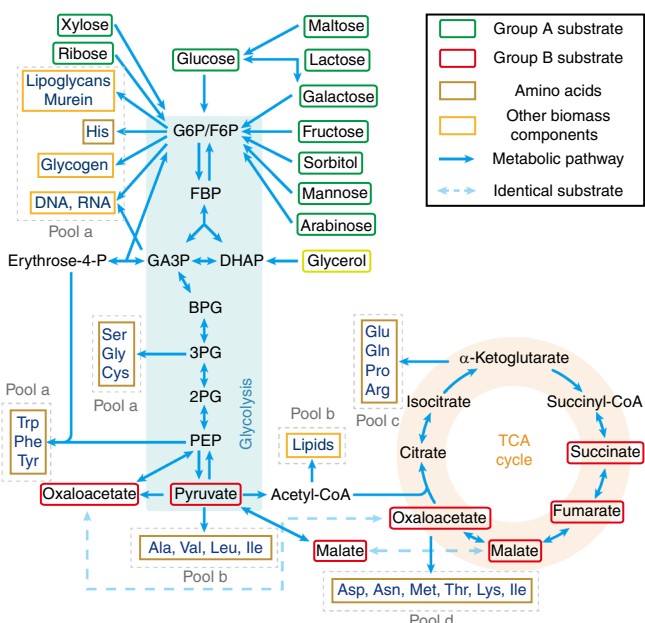

**Fig. 1** Metabolic network of carbon source utilization. Group A substrates (in green frames) can be simultaneously utilized with Group B substrates (red frames), whereas substrates paired from Group A usually display diauxie. Only the major pathways are shown. The precursors of biomass components (amino acids marked with light brown frames and other components marked with orange frames) are classified into Pools a–d (marked with gray dashed line frames). The enzyme for the interconversion between Glucose 6-phosphate (G6P) and fructose 6-phosphate (F6P) is very efficient (Supplementary Table 1), so we approximate G6P/F6P as a single node for convenience. All Group A carbon sources enter the metabolic network through G6P/F6P, while Group B carbon sources enter the metabolic network from different points after glycolysis. Glycerol enters from the upper part of glycolysis but not G6P/F6P, thus we classify glycerol as a quasi-Group A carbon source (see Supplementary Note 1.4 for details)

### Coarse-graining the metabolic network.
Note that the carbon sources from Group A converge to the node (G6P/F6P) before entering various pools, while the carbon sources from Group B can take other routes (Fig. 1). In fact, the metabolic network shown in Fig. 1 can be coarse-grained (see Methods) into a model shown in Fig. 2a, in which nodes $A1$ and $A2$ represent carbon sources from Group A, node $B$ a source from Group B, and Pools 1 and 2 are some combinations of the four pools in Fig. 1. In other words, Fig. 2a is topologically equivalent to Fig. 1 as far as the carbon flux is concerned. Each coarse-grained arrow (which can contain several metabolic steps) carries a carbon flux $J$ and is characterized by two quantities: the total enzyme cost $\Phi$ dedicated to carry the flux and a parameter $\kappa$ so that $J = \Phi \cdot \kappa$.

### Origin of diauxie for carbon sources in Group A.
Let us first consider the case in which both carbon sources are from Group A. We proceed to solve the simple model of Fig. 2a with two sources $A1$ and $A2$ ($[B] = 0$), using the optimal protein allocation hypothesis[18,20,21,25], which maximizes the enzyme utilization efficiency.

In Fig. 2a, all enzymes that carry and digest nutrient $Ai$ ($i = 1, 2$) into node $M$ are simplified to a single effective enzyme $E_{Ai}$ of cost $\Phi_{Ai}$ (see Supplementary Note 1.2 for details). The carbon flux to the precursor pools from source $Ai$ is proportional to $\Phi_{Ai}$. We take the Michaelis–Menten form (see Supplementary Note 1.2 for details): $J_{Ai} = \Phi_{Ai} \cdot \kappa_{Ai}$, where $\kappa_{Ai} = k_{Ai} \cdot \frac{[Ai]}{[Ai] + K_{Ai}}$ (denoted as the

substrate quality). $[Ai]$ is the concentration of $Ai$. For the subsystem consist of $A1$, $A2$ and node $M$, $J_{tot} = \Phi_{A1} \cdot \kappa_{A1} + \Phi_{A2} \cdot \kappa_{A2}$ and $\Phi_{tot} = \Phi_{A1} + \Phi_{A2}$. We define the efficiency of a pathway by the flux delivered per total enzyme cost[25,26]:

$$\varepsilon \equiv \frac{J_{tot}}{\Phi_{tot}}. \qquad (1)$$

The efficiency to deliver carbon flux from the two sources $A1$ and $A2$ to node $M$ is then $\varepsilon = \frac{\Phi_{A1} \cdot \kappa_{A1} + \Phi_{A2} \cdot \kappa_{A2}}{\Phi_{A1} + \Phi_{A2}}$. If $\kappa_{A1} > \kappa_{A2}$, that is, if the substrate quality of $A1$ is better than that of $A2$, then $\varepsilon = \kappa_{A1} - \frac{\Phi_{A2} \cdot (\kappa_{A1} - \kappa_{A2})}{\Phi_{A1} + \Phi_{A2}} \leq \kappa_{A1}$. It is easy to see that the optimal solution (maximum efficiency) is $\Phi_{A2} = 0$. This means that the cell expresses only the enzyme for $A1$ and thus utilizes only $A1$. Conversely, if $\kappa_{A1} < \kappa_{A2}$, $\Phi_{A1} = 0$ is optimal and the cell would utilize $A2$ only. In either case, optimal growth would imply that cells only consume the preferable carbon source, which corresponds to the case of diauxie[2,3,6,9,11].

In the above coarse-grained model, the enzyme efficiency of the carbon source $Ai$ is lump summed in a single effective parameter $\kappa_{Ai}$. In practice, there are intermediate nodes and enzymes along the pathway as depicted in Fig. 2b, and more elaborate calculations taking into account the intermediate steps are needed to evaluate the pathway efficiency. Note that Fig. 2b is rather generic in representing a part of the metabolic pathway under consideration. $X$ and $Y$ can represent carbon sources coming from Group A and/or Group B, $M$ represents the convergent node of the two sources under consideration and Pool z represents the precursor pool under consideration. We now proceed to calculate and compare the efficiencies of the two branches: $X \to M$ and $Y \to M$. Using the branch $X \to M$ as the example, denote $E_X^j$ the enzymes (of protein cost $\Phi_X^j$) catalyzing the intermediate nodes $m_X^j$ $(j = 1 \sim N_X)$ (Fig. 2b). Define $\Phi_X^b \equiv \Phi_X + \sum_{j=1}^{N_X} \Phi_X^j$, which is the total protein cost for enzymes dedicated to the branch. The pathway efficiency for the branch $X \to M$ is then $\varepsilon_{X \to M} = J_{X \to M}/\Phi_X^b$, where $J_{X \to M}$ is the carbon flux from $X$ to $M$. Assuming that the flux is conserved in each step along the branch, $J_{X \to M} = \Phi_X \cdot \kappa_X = \Phi_X^j \cdot \kappa_X^j$ $(j = 1 \sim N_X)$, where $\kappa_X^j = k_X^j \cdot \frac{[m_X^j]}{[m_X^j] + K_X^j} \approx k_X^j$ is the substrate quality of $m_X^j$ (Supplementary Note 1.2) and the last approximation is valid with $[m_X^j] \geq K_X^j$ which is generally true in bacteria[35,36] and which also maximizes the flux with a given enzyme cost (Supplementary Note 1.5). It is then easy to see that

$$\varepsilon_{X \to M} = \frac{1}{1/\kappa_X + \sum_{j=1}^{N_X} 1/\kappa_X^j} \approx \frac{1}{1/\kappa_X + \sum_{j=1}^{N_X} 1/k_X^j}. \qquad (2)$$

For $\varepsilon_{X \to M} > \varepsilon_{Y \to M}$, the optimal solution is $\Phi_Y = \Phi_Y^j = 0$; for $\varepsilon_{X \to M} < \varepsilon_{Y \to M}$, the optimal solution is $\Phi_X = \Phi_X^j = 0$ (see Supplementary Note 2.2 for details). Only the nutrient with higher branch efficiency is utilized to supply the convergent node $M$ and thus Pool z (Fig. 2b).

When $X$ and $Y$ both come from Group A, the convergent node $M$ resides upstream to all precursor pools (Fig. 1 and Supplementary Fig. 1c). Only the nutrient with higher efficiency is being utilized to supply all precursor pools. When the preferred nutrient is exhausted, the cell switch to the other less favorable nutrient. The actual switching point could depend on the concentrations of both nutrients. Note that the branch efficiency $\varepsilon_{Ai \to M}$ (Eq. (2)) depends on the concentration of the nutrient $[Ai]$ through the substrate quality $\kappa_{Ai} = k_{Ai} \cdot \frac{[Ai]}{[Ai] + K_{Ai}}$. Thus, only with

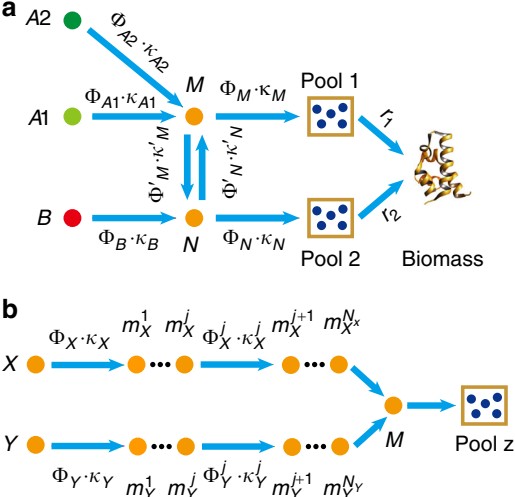

**Fig. 2** Topology of the metabolic network. **a** Coarse-grained model of the metabolic network. Group A carbon sources merge to a common node $M$ before reaching precursor pools. Group B sources can supply some precursor pools from other routes. **b** Topology of the part of the metabolic network connecting two carbon sources to a precursor pool. The two carbon sources $X$ and $Y$ (from Group A and/or Group B) reach a common node $M$ through multiple intermediate nodes (metabolites) $m_X^j$ and $m_Y^j$ along their respective pathway, after which the flux is diverted to Pool z

saturating concentrations, one can have an absolute ranking of the nutrient quality. For concentration $[Ai] < K_{Ai}$, $\varepsilon_{Ai \to M}([Ai])$ drops fast with $[Ai]$. Theoretically, when the concentration of the originally preferred nutrient ($A1$) drops to a point $[A1]_T$ such that $\varepsilon_{A1 \to M}([A1]_T) = \varepsilon_{A2 \to M}([A2])$, the other nutrient ($A2$) becomes preferrable, and the cell may switch to $A2$ at this point. This gives $[A1]_T = \frac{\delta \cdot [A2]}{\Delta + [A2]}$, where $\delta$ and $\Delta$ are constant (see Supplementary Note 2.3 for details). For $[A2] \ll \Delta$, the turning point is reduced to $[A1]_T = \frac{\delta}{\Delta} \cdot [A2]$, a form of ratio sensing. Indeed, ratio sensing was recently observed in the budding yeast *Saccharomyces cerevisiae* cultured in glucose-galactose mixed medium[33], and the experimental results agree well with the turning point equation derived above (see Supplementary Fig. 2 and Supplementary Note 2.3 for details).

**Co-utilization of carbon sources.** The diauxic growth is due to the topology of the metabolic network, in which Group A sources converge to a common node (G6P/F6P) before diverting to various precursor pools (Figs. 1, 2a and Supplementary Fig. 1c). The situation is different if the two mixed carbon sources are from Groups A and B, respectively (denoted as A + B). (Some combinations of two Group B sources also fall into this category and can be analyzed similarly; see Supplementary Fig. 3d) Group B sources can directly supply some precursor pools without going through the common node (G6P/F6P) (Fig. 1). The topologies of the metabolic network in the cases of A + B are exemplified in Supplementary Fig. 3. All A + B cases can be mapped to a common coarse-grained model depicted in Supplementary Fig. 1d (which is also Fig. 2a with only one of the A sources present), although the actual position of nodes $M$ and $N$ in the metabolic network, and the contents of Pools 1 and 2 may depend on specific cases. As obvious from Fig. 2a, source $A$ or $B$ alone could in principle supply all precursor pools. However, because of the location of the precursor pools relative to the sources, it may be more economical for one pool to draw carbon flux from one source and the other from the other source.

To determine which of the two carbon sources should supply which pool(s), we apply branch efficiency analysis (see Supplementary Note 3 for more details). For Pool 1, we compare the efficiency of $A$ and $B$ in supplying flux to node $M$; while for Pool 2 to node $N$. The criteria are simply:

$$\text{Pool 1 is supplied by} \begin{cases} A, & \text{if } \varepsilon_{A\to M} > \varepsilon_{B\to M} \\ B, & \text{if } \varepsilon_{A\to M} < \varepsilon_{B\to M} \end{cases}. \quad (3)$$

$$\text{Pool 2 is supplied by} \begin{cases} A, & \text{if } \varepsilon_{A\to N} > \varepsilon_{B\to N} \\ B, & \text{if } \varepsilon_{A\to N} < \varepsilon_{B\to N} \end{cases}. \quad (4)$$

It is easy to see from inequalities (3) and (4) that if the following condition is met

$$1/\kappa_B - 1/\kappa'_M < 1/\kappa_A < 1/\kappa_B + 1/\kappa'_N, \quad (5)$$

then $A$ supplies Pool 1 and $B$ supplies Pool 2—the two carbon sources are simultaneously consumed. In reality, there are multiple intermediate nodes between the $M$-$N$ interconversion (Figs. 1, 2a and Supplementary Fig. 1d). Similar to Eq. (2), $1/\kappa'_M$ and $1/\kappa'_N$ here actually represent summations of all intermediate terms between $M$ and $N$ in the metabolic network.

**Pools suppliers in the mixed carbon sources**. In order to apply the above analysis to the real case, we collected the available data for metabolic enzymes of *E. coli* from the literature (Supplementary Table 1). We calculated the branch efficiencies of different carbon sources to the metabolites F6P, GA3P, 3PG, PEP, pyruvate and oxaloacetate (see Supplementary Note 4.1 for details), which correspond to the nodes $M$ or $N$ in the simplified network of Fig. 2a for Pools a1–d (Fig. 1 and Supplementary Fig. 3). The results are shown in Supplementary Table 2. Then using the criteria Eqs. (3) and (4), one could evaluate the carbon source supplier(s) of all pools (Supplementary Table 3). Note that Pool c is supplied by both the suppliers of Pools b and d owing to the effect of converged flux (see Supplementary Notes 4.1–4.2 for details).

However, in practice, the suppliers of Pools b-d can be different from the above evaluation due to energy production in the TCA cycle. Specifically, when oxaloacetate flow through a TCA cycle back to itself, it generates fixed amount of energy[31], with half of the carbon atoms replaced by those coming from pyruvate (see Supplementary Figs. 4b-c and Supplementary Notes 4.3-4.4 for details). By collecting relevant energy production data from literatures[37], we quantitatively analyzed the influence of this effect, and obtained the optimal carbon source supplier(s) of each precursor pool for *E. coli* under aerobic growth in various combinations of source mixture (see Supplementary Notes 4.3–4.5 for details). The results are shown in Supplementary Table 4.

**Comparison with experiments**. To test these predictions (Supplementary Table 4), we use $^{13}$C isotope labeling methods to trace the carbon source(s) of each precursor pool (see Methods for details). Specifically, we cultured *E. coli* to the steady state in a mixture of two carbon sources with one source being labeled with $^{13}$C. We then measured the $^{13}$C labeling percentage of amino acids and obtained the percentage of each carbon source in supplying the synthesis of each amino acid. To ensure reliability, we obtained and compared our experimental results using two types of fragment in mass spec raw data: M-57 and M-85 (Supplementary Fig. 5) (see Methods for details).

We first examined the A + B cases (Group A source: glucose, lactose, fructose, glycerol; Group B source: pyruvate, succinate, fumarate, malate). Overall, the experimental results showed excellent agreement with our predictions (Fig. 3). A number of features are worth noting. Just as the model predicted

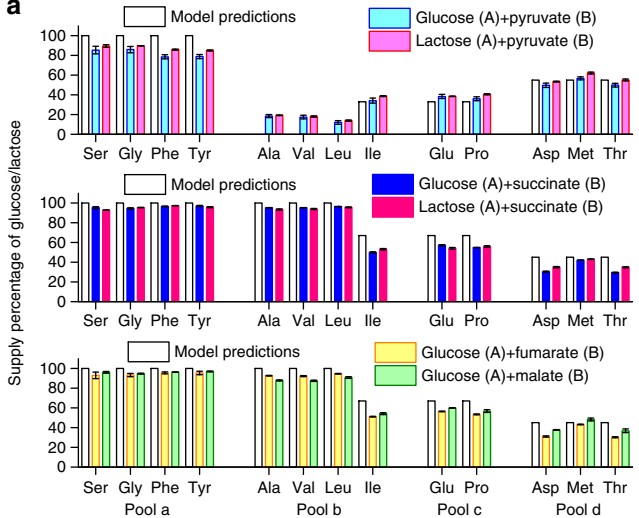

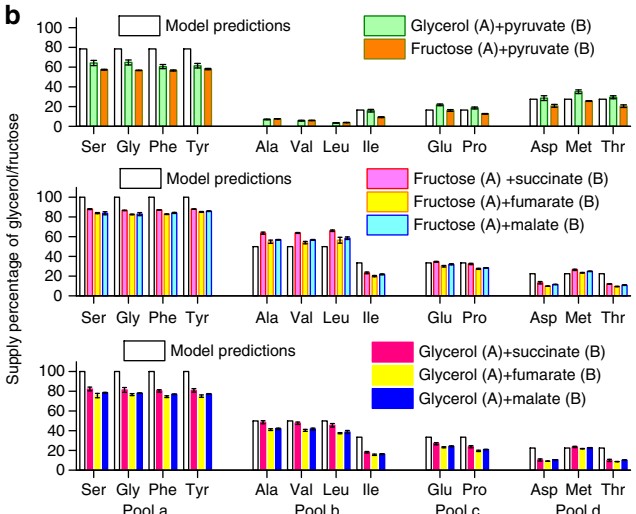

**Fig. 3** Suppliers of precursor pools in A + B cases. Vertical axes are the percentages of the carbon atoms from the first of the two sources indicated. Model predictions (in hollow bars, see Supplementary Table 4) are shown together with experimental results (in color bars). The source supplier of representative amino acids in Pools a–d was measured using $^{13}$C labeling (raw data from M-57 fragment; see Methods for details). Error bars represent standard deviations. Source data are provided as a Source Data file. **a** Glucose or lactose mixed with a Group B carbon source. **b** Fructose or glycerol mixed with a Group B carbon source

(Supplementary Table 4), two patterns of the carbon source partition (Fig. 3a, b) were observed depending on which A source was used. Glucose and lactose are both highly preferable carbon sources for *E. coli*, both supporting large growth rates (Supplementary Table 5); their supply patterns look almost the same (Fig. 3a). Fructose, glycerol, maltose and galactose are less preferable Group A sources with lower growth rates (Supplementary Table 5), and they showed very similar supply patterns when mixed with the same Group B source (Fig. 3b, Supplementary Fig. 6a). Group B sources succinate, fumarate and malate showed similar supply patterns when mixed with a same Group A source (Figs. 3a, b, 4a & Supplementary Fig. 6a). There was a noticeable systematic discrepancy between the experimental results and the model predictions for Pool a. This may be due to the fact that microbes reserve a portion of gluconeogenesis enzymes preparing for potential changing environment (see also Supplementary Notes 4.8–4.9).

Next, as our model can calculate carbon source utilization and partition in any combinations of sources, we performed experiments for B + B cases (pyruvate mixed with succinate, fumarate or malate; succinate mixed with malate). In agreement with the model prediction, these B + B cases showed co-utilization and the measured carbon supply percentages quantitatively agree with model predictions (Fig. 4).

## Discussion

The diauxie versus co-utilization puzzle can be understood from the topology of the metabolic network. This can be illustrated with the coarse-grained model shown in Fig. 2a (see also Supplementary Figs. 1b–d). The sources of Group A go through a common node before delivered to various precursor pools, and the most efficient source wins[22]. It has been observed that there is a hierarchy among Group A sources ranked according to the single-source growth rate, and when two or more sources are present the bacteria usually use the one with highest growth rate[6,34]. This is a natural consequence of our theory. A higher growth rate commonly implies higher enzyme utilization efficiency and thus a higher priority to be utilized. Other than the *lac* operon, questions remain as how this priority is implemented molecularly. It has been known that in many cases the catabolite repression is not complete and that this may depend on whether the carbon sources belong to the type of Phosphotransferase system (PTS)[10,13], highlighting the potential constraints, trade-offs and/or costs of implementing a prefect optimal solution. We have mixed glucose (a PTS sugar) with both PTS sugar fructose and non-PTS sugars maltose and glycerol, which all belong to the A + A cases. We found that while glucose showed almost perfect inhibition to the two non-PTS sugars (all precursor pools were about 100% supplied by glucose), its inhibition to the other PTS sugar fructose was not complete (glucose supplied ~83% precursor pools) (Supplementary Fig. 6b).

When Group B source is present along with Group A source, it can take a shortcut to reach some of the precursor pools (Fig. 2a, Supplementary Fig. 1d) and can be more efficient to supply these pools. Some combinations of two Group B sources also fall into this category and thus can be co-utilized. In these cases, our experimental results quantitatively agree with our model predictions. As can be seen from Figs. 3 and 4, despite the various possible combinations of carbon sources, the partition of the sources among the pools fall into a few patterns. This is due to the fact that these partitions are largely determined by the topology of the metabolic network and thus are quantized. This property also relaxes the requirements of accurate enzyme parameters in determining the pool suppliers. To test the robustness of the model predictions with respect to the errors/uncertainties of the parameters extracted/estimated from the literature, we carried out a detailed analysis (Supplementary Note 4.9 and Supplementary Tables 6 and 7). The analysis showed that for any mixture of two carbon sources and for arbitrary choice of parameters, only a very few (no more than 4) partition patterns of the sources are qualitatively similar to the experimental result. Model predictions using the nominal parameter values from the literature quantitatively and consistently agree with all the experimental patterns for all the combinations of carbon sources we tested. Conversely, in order to produce a pattern that is qualitatively similar but quantitatively different from the experimental one, a very large deviation from at least one nominal value is necessary.

The present work deals with relatively stable growth conditions and the simple exponential growth behavior. In this case, there is a body of experimental evidence for optimal protein allocation[18–24,27,38]. Furthermore, our model relies only on the assumption that microbes optimize enzyme utilization efficiency,

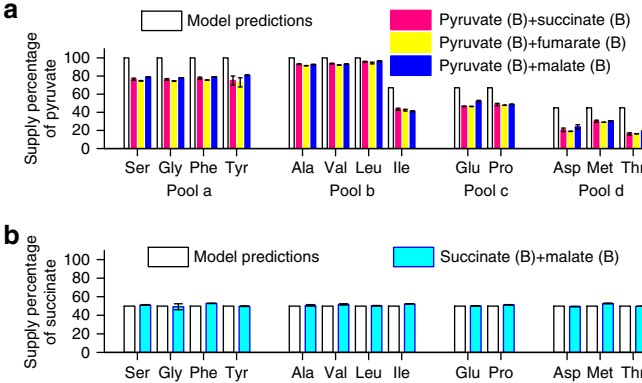

**Fig. 4** Suppliers of precursor pools in B + B cases. Vertical axes are the percentages of the carbon atoms from the first of the two sources indicated. Model predictions (in hollow bars, see Supplementary Table 4) are shown together with experimental results (in color bars). The source supplier of representative amino acids in Pools a–d was measured using $^{13}$C labeling (raw data from M-57 fragment; see Methods for details). Error bars represent standard deviations. Source data are provided as a Source Data file. **a** Pyruvate mixed with another Group B carbon source. **b** Succinate mixed with malate. In this case, the branch efficiencies of the two sources are about the same

so it may also be applicable to suboptimal growth cases[27,38]. The environment the microbes face can be highly variable and uncertain. Their long-term fitness of the population may not simply be determined only by the growth rate of individual cells in the exponential phase, but a result of trade-offs that best adapt to the changing environment. Strategies such as bet hedging, memory of the past and anticipation of the future are found to exist in microorganisms[39–48].

Finally, from theoretical aspects, our analysis framework is broadly applicable to more complex regulations in metabolic networks such as reversible reactions, allosteric enzymes, metabolites inhibitions, etc. (see Supplementary Notes 5–7 for details). However, there are cases, such as bi-substrates transporters or enzymes (e.g., glucose transporters in *E. coli* can co-transport mannose[49]), for which specific care needs to be taken (see Supplementary Notes 6–7 for details). In practice, the nutrient uptake strategy or eating habit of a microbe is shaped by its environmental history. While the phenomena of diauxie versus co-utilization are widely spread in microbes, there are bound to be variations and exceptions. For example, certain microbes may have different hierarchies of preferable carbon sources[4]. It is still a great challenge to understand in quantitative frameworks how cells and population behave and evolve in different and changing environments.

## Methods

**Coarse graining methods**. Coarse graining of the metabolic network is done in such a way as to preserve the network topology but grouping metabolites, enzymes and pathways into single representative nodes and corresponding effective enzymes. In particular, a linear pathway is lump summed into two nodes (start and end) connecting with a single effective enzyme.

**Strain**. The strain used in this study is *E. coli* K-12 strain NCM3722.

*Growth medium*: Most of the cultures were based on the M9 minimal medium (42 mM Na$_2$HPO$_4$, 22 mM KH$_2$PO$_4$, 8.5 mM NaCl, 18.7 mM NH$_4$Cl, 2 mM MgSO$_4$, 0.1 mM CaCl$_2$), and supplemented with one or two types of carbon sources. For the carbon sources in each medium, the following concentrations were applied: 0.4% (w/v) glucose, 0.4% (w/v) lactose, 0.4% (v/v) glycerol, 20 mM fructose, 20 mM fructose, 20 mM maltose, 20 mM pyruvate, 15 mM succinate, 20 mM fumarate, and 20 mM malate.

*Batch culture growth*: The batch cultures were performed either in the 37 °C incubator shaker shaking at 250 rpm or in the microplate reader, which holds the temperature at 37 °C and shakes at 900 rpm. The culture volume was 200 μL in 96-well plates, 1 mL in 5 mL round-bottom tubes or 50 mL in 100 mL flasks. Every batch culture was performed as described below. Single colony from the LB agar plate was first inoculated into 50 mL LB medium and cultured overnight. Then, 0.5 mL overnight culture was inoculated into 50 mL LB medium and cultured for 2 h. Cells were centrifuged at 4000 rpm for 2 min, and the cell pellets were diluted to $OD_{600} = 0.001$ in the culture medium (M9 medium supplemented with various carbon sources). Then, the medium was cultured in microplate reader (measuring growth rate) or incubator shaker (isotope labeling).

*Growth rate measurement*: Each well of the 96-well plate was covered with a 200-μL culture medium ($OD_{600} = 0.001$). Cells were cultured at 37 °C in the microplate reader shaking at 720 rpm with a 2-mm diameter. The microplate reader measured the $OD_{600}$ of each well at an interval of 5 min for 20 h. Growth rate $\lambda$ was measured as the multiplicative inverse of doubling time:

$$\lambda = \frac{d \log_2 OD_{600}}{dt}. \tag{6}$$

The interval of 2 h with the maximum Pearson correlation coefficient was defined to be in the exponential phase and the slope of this interval was the growth rate. For some cultures, the $OD_{600}$ remains constant during the first 20 h. Thus the record time was extended to 72 h for these cultures.

*Isotope labeling*: The following $^{13}C$ carbon sources were applied in the isotope labeling experiments: glucose (Product code: CLM-1396; Cambridge Isotope Laboratories, Inc.), fructose (Product number: 587621; Sigma-Aldrich) and glycerol (Product number: 489476; Sigma-Aldrich) (Group A); pyruvate (Product code: CLM-2440; Cambridge Isotope Laboratories, Inc.) and succinate (Product number: 491985; Sigma-Aldrich) (Group B). In an isotope labeling experiment, there are two types of carbon sources: one is uniformly labeled with $^{13}C$, while the other one is not labeled. In every experiment, 1 mL $^{13}C$-labeled culture medium ($OD_{600} = 0.001$) was inoculated into the 5 mL round-bottom tube and cultured in the incubator shaker until the $OD_{600} = 0.150$ to 0.250 (7–8 generations). Three independent experiments (with numerous distinct cells) were carried out for each combination of mixed carbon sources.

*Extraction and derivatization of amino acids*[50]: Cells labeled by $^{13}C$ were harvested by centrifuging for 3 min at 12,000 rpm. The cell pellets were washed with 1 mL PBS and centrifuged for 2 min at 12,000 rpm twice, and then resuspended in 200 μL of 6 M HCl. The resuspended cells were transferred into sealed 1.5 mL tubes and hydrolyzed for 20 h at 105 °C. The cell hydrolysate was dried at 65 °C under the fume hood. The dry hydrolysate was resuspended with 40 μL N,N-Dimethylformamide and 20 μL N-tert-butyldimethylsilyl-N-methyltrifluoroacetamide and heated at 85 °C for 1 h so that the amino acids were derivatized (structure shown in Supplementary Fig. 7b). The solution of mixed derivatized amino acids was filtered with 13 mm syringe filter with 0.2 μm membrane.

*GC-MS setup*: GC-MS analysis was carried out using the Hybrid Quadrupole-Orbitrap GC-MS/MS System (Q Exactive GC, ThermoFisher). The injected sample volume was 1 μL at a carrier gas flow of 1.200 mL/min helium with a split ratio of 1:4.2. The oven temperature was initially set at 150 °C and maintained for 2 min, raised to 180 °C at 5 °C/min and immediately to 260 °C at 10 °C/min and maintained for 8 min, and then raised to 350 °C/min and maintained for 5 min. The ionization mode was set as electron impact ionization. The ion source temperature was set at 230 °C. The MS transfer line temperature was set at 250 °C. The scan range was 50.0 to 650.0 $m/z$ with a resolution at 60,000. The MS was tuned to 414.0 $m/z$.

*GC-MS analysis of $^{13}C$ labeled derivatized amino acids*: The derivatized amino acids was analyzed by GC-MS with the setup described above. Different kinds of derivatized amino acids in one sample were separated in the gas chromatography (GC) according to their different retention time. The amino acids were fragmented during ionization, forming different kinds of fragments (Supplementary Fig. 7b). Different kinds of fragment of the same derivatized amino acid and their relative abundance (Supplementary Fig. 7b–c) were analyzed by mass spectrometry (MS). The labeling percentage of a certain amino acid can be inferred from the labeling percentage of its fragments. Thermo Xcalibur4.0 was used to view and process the GC-MS data. According to the relative retention time in the chromatogram (Supplementary Fig. 7a) and the corresponding mass spectrum (Supplementary Fig. 7b), 13 kinds of amino acid were detected. The integrated mass spectrum over the full peak range of every derivatized amino acid was obtained to calculate the $^{13}C$ labeling percentage[51]. For a typical derivatized amino acid, 5 types of fragment, M-15, M-57, M-85, M-159 and f302 (Supplementary Fig. 7b), were detected. M-57 denotes that this fragment weighs 57 daltons less than the corresponding derivatized amino acids, and the same goes for M-15, M-85 and M-159. f302 denotes a fragment of weights 302 daltons. For a certain fragment containing N carbon atoms from the underivatized amino acids (the natural form of concern), there were $(N + 1)$ kinds of mass isotopomer incorporating $0 \sim N$ $^{13}C$ respectively. $I_i$ denoted the intensity of mass isotopomer that had $i$ $^{13}C$ and $N - i$

$^{12}C$. The $^{13}C$ labeling percentage $\zeta$ of this fragment was calculated as follows:

$$\zeta = \frac{\sum_{i=1}^{N} i \cdot I_i}{N \cdot \sum_{i=1}^{N} I_i}. \tag{7}$$

*Amino acids $^{13}C$ labeling data of different fragments*: There are five types of fragments (M-15, M-57, M-85, M-159 and f302) formed during the ionization of amino acids. With each type of raw data, we can calculate (with Eq. (7)) a set of $^{13}C$ labeling percentages for amino acids. According to the molecular structure, these results can be classified into the following three categories: using (1) M-15/M-57; (2) M-85/M-159; and (3) f302. Only in the first category, there is no carbon atom loss during the fragmentation of the amino acids of concern (Supplementary Fig. 7). This means that M-15/M-57 reflects the exact $^{13}C$ labeling percentages of amino acids. Yet, the signal intensity of M-15 is faint, thus M-57 is the best choice among the fragments. However, in practice, Leu M-57 and Ile M-57 fragments share the same mass as that of f302 and thus are not applicable for analysis. Consequently, we used Leu M-15 and Ile M-15 to calculate the $^{13}C$ labeling percentage of Leu and Ile, and M-57 data for other amino acids throughout our manuscript and Supplementary Information unless otherwise specified. In the second category, the carboxylic carbon atom in an amino acid was lost during the fragmentation (Supplementary Fig. 7). Thus M-85/M-159 data can reflect the $^{13}C$ labeling percentages for a majority yet not all carbon atoms in amino acids, and indeed M-85 and M57 data show a very good agreement with each other (Supplementary Fig. 5). In the third category, all carbon atoms in the side chain of an amino acid (a considerable proportion) were lost during fragmentation (Supplementary Fig. 7). As a result, we did not use f302 data for calculation in this study.

**Reporting summary**. Further information on experimental design is available in the Nature Research Reporting Summary linked to this article.

## Code availability

This paper does not involve computer code. Built-in functions of Origin (v9) were used for curve fitting in Supplementary Fig. 2.

## Data availability

The data that support the findings of this study are available from the corresponding author (C.T.) upon request. The source data underlying Figs. 3–4, Supplementary Figs. 5–6, Supplementary Note 4.7 and Supplementary Table 5 are provided with the paper.

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

## Acknowledgements

We thank Yiping Wang for providing us the strain NCM3722, and Terence Hwa, Yuan Yuan, Haoyuan Sun, Xiaohui Liu, Yanjun Li, Shaoqi Zhu, and Yimiao Qu for helpful discussions. This work was supported by Chinese Ministry of Science and Technology (2015CB910300) and National Natural Science Foundation of China (91430217).

## Author contributions

X.W. and C.T. conceived and designed the project, developed the model, designed the experiments, and wrote the paper. X.W. carried out the theoretical analysis. K.X. carried out the experiments. X.W. and K.X. analyzed the experimental data. C.T. supervised the whole project. X.Y. co-supervised the experiments.

## Additional information

**Competing interests:** The authors declare no competing interests.

