## [Peer Review File · Nature Communications]

Reviewers' comments:

Reviewer #1 (Remarks to the Author):

Comment paper Tang.

This paper provides theory that forms an explanation for catabolite repression and for the co-consumption of carbon sources from an optimisation perspective. For a theoretical paper in Nature Communications, one would expect significant conceptual novelty. However, this work extends a previously developed coarse-grained resource allocation theory from the Hwa group, but even worse, completely ignores existing and more advanced theory that does not require some of the simplifications used here. This theory, published independently by Muller et al (JTB 2014) and Wortel et al (FEBS J 2014), shows that when flux is optimised under a fixed total protein constraint, then the optimal state will be a flux through a single elementary flux mode (EFM). This outcome is irrespective of the enzyme kinetics and network topology, and is therefore much more general than the specific case of glucose and galactose presented here. The here proposed Hwa-like theory ignores kinetics and fixes saturation, in a poorly-justified way (see specific comment 1). In fact, the results of this submitted theory are all specific cases of the EFM-based theory. Also the conditions for co-consumption have a topological underpinning, as it appears that only those carbon sources are co-consumed that can form a single EFM towards biomass.

Prof Tang should be aware of this body of theory because I discussed this with him at the Barcelona ICSB conference and I send him the Wortel paper – of which I am a co-author. It is therefore regrettable that this work is not mentioned.

Now the coarse-grained theory does not require extensive kinetic data, as does the EFM theory, to provide quantitative predictions, because it is coarse grained. In that sense this theory may be a useful coarse graining of existing theory, but then the usefulness must be demonstrated and informed by quantitative experiments (see comment 2).

Specific comments

- 1) On line 87 of the Main Text the authors state that the carbon flux to the precursor pools from a source A_i occurs in one reaction step which has the irreversible Michaelis-Menten kinetics: $J_{A_i} = \varphi_{A_i} \kappa_{A_i}$, where $\kappa = k \frac{A_i}{A_i + K}$. Assuming irreversible Michaelis-Menten kinetics is a very strong simplification, since in reality we need to account for reversibilities, allosteric feedback, product inhibition etcetera. In the Supplementary Information this simplification is not further justified.

The simplification that the precursor pool supply occurs in one step is somewhat clarified in the SI Section 6, where a two-step scenario is considered. The authors then get an extra inequality: $\varphi_R \kappa_t \leq \phi_M \kappa_M \leq \varphi_{A_1} \kappa_{A_1}$ and then state that the growth rate λ is maximized only when all the inequalities are set equal and that it is clear that λ is only maximized when the intermediate enzymes are almost saturated: $\kappa_i = k_i \frac{S_i}{S_i + K_i} \approx k_i$. They admit that the real situation could be more complicated, but state: “however, **there is no causal link** between κ_M and κ_{A_1} , so we can simply regard κ_M as a constant”. And this is where the authors are definitely wrong, because the absence of the causal link is only true in their simplified irreversible Michaelis-Menten kinetics. Even if we only add the next layer of complexity, high intermediate metabolite concentrations will speed up reactions that take this metabolite as a substrate while it slows down reactions that produce this metabolite

(due to the activity of the reverse reaction). This creates a link such that the variables κ cannot be seen as independent constants and enzyme saturations can thus not be kept fixed. Moreover, the argument for taking a one-step supply is invalidated.

The argument of highly saturated enzymes is used again on line 121 of the Main Text and 170 of the SI.

- 2) Starting from line 270 in the SI, the authors state that there is currently no experimental data to compare with the predictions in Supplementary Table 3, which is a prediction of which resource pool is supplied by which carbon source. Since this is the most predictive statement of the whole work, this demands an experiment. It could for example be tested if the reactions connecting two of the pools are actually switched off when growing on multiple substrates.
- 3) The authors dedicate a section in the main text to data from Escalante-Chong et al. This data shows heterogeneity in a yeast population that switches from galactose to glucose. Escalante-Chong et al. have already concluded that *S. cerevisiae* decides which carbon source to consume depending on the ratio between the sources. They find an almost straight 'switching line' when galactose (%) is plotted versus glucose (%). That the authors can fit this straight line using 5 parameters can hardly be called 'remarkable'.

The remarkable aspect of this experimental dataset is the heterogeneity in the population. If the authors could explain this from an optimization perspective, then it would add something valuable to the existing theory of biological optimization.

Reviewer #2 (Remarks to the Author):

When grown on a medium containing two carbon substrates, microbes often take up just one, resulting in the famous phenomenon of diauxie. Based on the fact that the elected substrate is usually the one that provides the faster growth, many authors have speculated that this behavior is in fact optimal: assuming that all metabolic precursors required for growth can be derived from either carbon substrate, it makes sense to just consume the substrate that does so with the smallest investment. Previously, authors typically assumed that the "currency" of the investment was "energy"; in the light of the recent work of the Hwa group and others, it makes sense to replace this with "proteome fraction", but the argument stays the same.

While this old argument sounds plausible, it does not explain the basic fact that, in *E. coli* and other microbes, many substrate pairs are in fact co-utilized. Why doesn't the argument hold for these cases?

This manuscript goes beyond the old hypothesis by taking into account that the efficiency of each carbon substrate (in terms of investment of proteome) actually differs for each molecular precursor (e.g., different amino acids) required for growth. They show that, by taking this into account, it becomes clear that the optimal solution (the one maximizing the growth rate) then depends on the entry point of each substrate into the metabolic network, and on the topology of the network itself.

Many good ideas are obvious in retrospect. This is such an idea. The authors have convinced me that this argument indeed an important factor in the distinction between co-utilized and sequentially utilized substrate pairs.

An important aspect in favor of the theory is that it makes predictions that are rather easy to verify experimentally, and the authors do in fact provide such predictions (shown in Supp. Tables). It is a bit of a shame that the authors did not team up with experimentalists in order to verify these predictions, but I for one believe that publishing theoretical predictions before their experimental verification should become more common in biology -- this practice has served other disciplines (physics, chemistry) well for the past few centuries.

Some points that the authors should (be able to) address:

* In figure 1, glycerol is categorized into group B. It is known, however, that glycerol is not co-utilized with glucose or lactose. Can the model explain this? If the authors "predict" the utilization pattern for glucose + glycerol with their model (as in Supplementary Table 3), what do they find?

* The results in Fig. 5 show some correlation between the predictions and the actual uptake measurements, but I'm not too impressed by the predictions for mixture 2 and 3. Why and how do the authors decide that the predictions are "consistent" with the experimental results?

* Generally speaking, the manuscript is clearly written and the figures/illustrations have been carefully drawn. The Discussion section, however, should definitely be improved. Currently, it reads as an incoherent collection of statements and topics that had to be mentioned somewhere. For instance, what is the main message of the first paragraph? And what is the point of the paragraph starting with "Cell growth is a fundamental issue in biology.?" What does that sentence even mean?

* Here and there, I found the mathematical notation to be unnecessarily complex. In line 83, for instance, the indices within indices can easily be avoided, and the protein mass of the cell could simply be called M . Throughout, indices " A_i " could, I believe, simply be replaced with " i ". The point is: I field that the authors should try to minimize and simplify the mathematical expression in the main text, obviously without compromising the main message, in order to ensure that the paper is

accessible to the wide readership of this journal.

Reviewer #3 (Remarks to the Author):

In this manuscript, the authors aim to understand the two different carbon utilizing behaviors exhibited by bacteria growing on carbon mixtures: sequential utilization (diauxie) and simultaneous utilization (co-utilization). With mathematical analysis of coarse-grained metabolic models, they demonstrated that the two different behaviors originate from the principle of optimal protein resources allocation for maximum growth. The study excels at abstracting simple coarse-grained models (e.g., Fig. 2c and Fig. 4b) from the complicated metabolic network. These concise but biologically relevant models allow rigorous mathematical analysis for biological insights. This work is an example of first-class theoretical biology studies. There are however a few points that need to be clarified before being published, particularly about the comparisons between model predictions and experimental data. I also have some suggestions for improving the manuscript.

Major:

1. The authors mentioned a comparison between the theory and experiment with *Methylobacterium extorquens* AM1 (Page 13). I however couldn't locate such a comparison. It is not in the SI.
2. It is not clear what is exactly being compared in Fig. 5. How were the "experimental results" obtained? These numbers are not directly available in the reference 6. Were they calculated using Eqn. S43? But according to the text after Eqn. S43, the results from the equation are not meant to be "experimental results" because "they are plotted in Fig. 5 along with the experimental data" (Line 308 in the SI). What about the "model predictions"? Were they calculated utilizing the numbers in Line 281 and Line 282 in the SI? In sum, it seems that the main text and SI are contradictory about Fig. 5; and in any case the text about Fig. 5 is too brief.
3. In Eqns. 1 and 2, more explanations on the inequalities are needed. For example, doesn't flux balance always enforce equality in Eqn. 2? Is the inequality due to flux leakage? And the inequality in Eqn. 1 depends on how ϕ_{\max} is defined.

Minor:

4. While referring to the SI, instead of saying the same "see SI for details" through the text, the authors should provide more specific locations, e.g., see Sec. 5 of SI for details.
5. I find the use of "nutrient quality", "nutrient efficiency", and "substrate quality" confusing. They have the same units. Is it possible to use the same term for all of them? Along the same line, is it possible to use a single symbol for nutrient efficiency (epsilon) and nutrient quality (kappa)?
6. In Line 87, k_{Ai} and K_{Ai} are not defined.
7. In Line 129, k_M should be κ_M .
8. In Line 145, instead of using the subtly different k_{A1}^{A2} and K_{A1}^{A2} , consider use a different symbol other than k for them.
9. In Line 34 of SI, what is the physical/biological meaning of the coefficient α ?
10. In Line 139 of SI, ϕ_R should be ϕ_S .
11. In Line 172 of SI, k_M should be κ_M .
12. In Line 178 of SI, k_M should be κ_M .
13. In Line 189 of SI, it is a jump from κ_{Ai}^j in Eqn. S30 to k_{Ai}^j in Eqn. S32. The approximation should be explicitly indicated during the derivation.

Response to reviewers

(Reviewers' comments in black and our response in blue)

Reviewer #1:

This paper provides theory that forms an explanation for catabolite repression and for the co-consumption of carbon sources from an optimization perspective. For a theoretical paper in Nature Communications, one would expect significant conceptual novelty. However, this work extends a previously developed coarse-grained resource allocation theory from the Hwa group, but even worse, completely ignores existing and more advanced theory that does not require some of the simplifications used here. This theory, published independently by Muller et al (JTB 2014) and Wortel et al (FEBS J 2014), shows that when flux is optimised under a fixed total protein constraint, then the optimal state will be a flux through a single elementary flux mode (EFM). This outcome is irrespective of the enzyme kinetics and network topology, and is therefore much more general than the specific case of glucose and galactose presented here. The here proposed Hwa-like theory ignores kinetics and fixes saturation, in a poorly-justified way (see specific comment 1). In fact, the results of this submitted theory are all specific cases of the EFM-based theory. Also the conditions for co-consumption have a topological underpinning, as it appears that only those carbon sources are co-consumed that can form a single EFM towards biomass. Prof Tang should be aware of this body of theory because I discussed this with him at the Barcelona ICSB conference and I send him the Wortel paper – of which I am a co-author. It is therefore regrettable that this work is not mentioned.

Now the coarse-grained theory does not require extensive kinetic data, as does the EFM theory, to provide quantitative predictions, because it is coarse grained. In that sense this theory may be a useful coarse graining of existing theory, but then the usefulness must be demonstrated and informed by quantitative experiments (see comment 2).

Response: We thank the reviewer for both the critical and suggestive comments. We apologize for our negligence that we did not fully appreciate the two Elementary Flux Mode (EFM) papers (Muller et al (JTB 2014) and Wortel et al (FEBS J 2014)) at the time of our first submission. Indeed, we found these two EFM papers are very relevant to our study, and have now cited and stressed the significance of these two papers both in the manuscript and in the Supplemental Information.

We build our model framework based on optimal allocation of protein resources in microbes in combination with the topological structure of the metabolic network. The optimization principle has been studied for years by many authors including Hwa and those of the two papers, and so has the metabolic network. The major contribution of our study is that we recognized the important role of the metabolic network topology in determining the microbial growth behavior, with which we explained the diauxie versus co-utilization puzzle in the optimization framework.

Our study applies the similar logic of optimization principle as the two EFM papers, i.e. to maximize the enzyme utilization efficiency. However, the results of our study are not merely specific cases of the EFM-based theory. Taking the case of co-utilization (Fig. S1d) as an example, where A and B are two different carbon sources from Group A and Group B, respectively. Our theory can predict the optimal strategy of the microbes: using only A, using only B, or using both A and B. Furthermore, our theory can predict quantitatively the carbon supply percentage of precursor's pools (e.g. amino acids) from each carbon source. EFM-based theory, on the other hand, even if combining with the topological features of the metabolic network, is still unable to predict whether co-utilization is the optimal strategy or not. Contrary to this reviewer's expectation, the case that the two carbon sources (A and B) are co-consumed is not the only EFM solution. Let us analyze the EFMs in this case (the figure below, see also Fig. S1d) as below:

Here we follow the same analysis procedures in the original EFM paper (Schuster, S., & Hilgetag, C. (1994). On elementary flux modes in biochemical reaction systems at steady state. *Journal of Biological Systems*, 2, 165-182.). The reactions and nodes of concerns are listed in the following table.

Reaction nodes of the network	Reactions
S1=M; S2=N; S3=Pool1; S4=Pool2; S5=Biomass;	v1: A→M; v2: B→N; v3: M→Pool 1; v4: N→Pool 2; v5: M→N; v6: N→M; v7: r1*Pool1+ r2*Pool2→Biomass; (r1+r2=1) v8: Biomass→

The stoichiometry matrix is

$$N = \begin{pmatrix} 1 & 0 & -1 & 0 & -1 & 1 & 0 & 0 \\ 0 & 1 & 0 & -1 & 1 & -1 & 0 & 0 \\ 0 & 0 & 1 & 0 & 0 & 0 & -r_1 & 0 \\ 0 & 0 & 0 & 1 & 0 & 0 & -r_2 & 0 \\ 0 & 0 & 0 & 0 & 0 & 0 & 1 & -1 \end{pmatrix},$$

which results in three EFMs:

$$\begin{cases} V^{(1)} = (1 & 0 & r_1 & r_2 & r_2 & 0 & 1 & 1)^T \\ V^{(2)} = (0 & 1 & r_1 & r_2 & 0 & r_1 & 1 & 1)^T \\ V^{(3)} = (r_1 & r_2 & r_1 & r_2 & 0 & 0 & 1 & 1)^T \end{cases}.$$

Evidently, all $V^{(i)}$ ($i = 1-3$) satisfy both $N \cdot V^{(i)} = 0$ ($i = 1-3$) and the simplicity requirement that no $V^{(i)}$ ($i = 1-3$) is a non-negative linear combination of other elementary flux modes. Consequently, all $V^{(i)}$ ($i = 1-3$) are EFMs. The case that both A and B are consumed form one of the EFMs ($V^{(3)}$), yet there are another two EFMs: $V^{(1)}$ and $V^{(2)}$, corresponding to using only A and using only B, respectively. EFM-based theory shows that the optimal strategy is an EFM, yet cannot tell which one is the optimal strategy when there are three EFMs. Meanwhile, we find there are also exceptional cases that violate the EFM-based theory, with the optimal state not an EFM (see SI Sec. 6.5 for the exceptional cases).

Specific comments

1) On line 87 of the Main Text the authors state that the carbon flux to the precursor pools from a source A_i occurs in one reaction step which has the irreversible Michaelis-Menten kinetics: $J_{A_i} = \phi_{A_i} \cdot \kappa_{A_i}$, where $\kappa_{A_i} = k_{A_i} \cdot \frac{[A_i]}{[A_i] + K_{A_i}}$. Assuming irreversible Michaelis-Menten kinetics is a very strong simplification, since in reality we need to account for reversibilities, allosteric feedback, product inhibition etcetera. In the Supplementary Information this simplification is not further justified.

The simplification that the precursor pool supply occurs in one step is somewhat clarified in the SI Section 6, where a two-step scenario is considered. The authors then get an extra inequality: $\phi_R \cdot \kappa_i \leq \phi_M \cdot \kappa_M \leq \phi_{A1} \cdot \kappa_{A1}$ and then state that the growth rate λ is maximized only when all the inequalities are set equal and that it is clear that λ is only maximized when the intermediate enzymes are almost saturated: $\kappa_i = k_i \cdot \frac{s_i}{s_i + K_i} \approx k_i$.

They admit that the real situation could be more complicated, but state: "however, **there is no causal link** between κ_M and κ_{A1} , so we can simply regard κ_M as a constant". And this is where the authors are definitely wrong, because the absence of the causal link is only true in their simplified irreversible Michaelis-Menten kinetics. Even if we only add the next layer of complexity, high intermediate metabolite concentrations will speed up reactions that take this metabolite as a substrate while it slows down reactions that produce this metabolite. (due to the activity of the reverse reaction). This creates a link such that the variables κ_M cannot be seen as independent constants and enzyme saturations can thus not be kept fixed. Moreover, the argument for taking a one-step supply is invalidated.

The argument of highly saturated enzymes is used again on line 121 of the Main Text and

170 of the SI.

Response: We thank the reviewer for thoughtful considerations and suggestions. We have performed detailed analyses on reversible reactions, cooperative effect, product inhibition, and other complex cases and regulations of the metabolic network. The analyses show that in general these more complex regulations do not invalidate our model. We have added three sections in the Supplemental Information (SI Sec. 5-7) to specifically address this issue.

The statement “there is no causal link between κ_M and κ_{A1} ” in our original manuscript is inappropriate. We thank the reviewer for pointing it out. We have deleted this

sentence. As for the approximation $\kappa_i = k_i \cdot \frac{[S_i]}{[S_i] + K_i} \approx k_i$ (or the “saturated enzymes”

expression), our main goal is to estimate the value of κ_i . Recent studies (Bennett BD, Kimball EH, Gao M, Osterhout R, Van Dien SJ, Rabinowitz JD., 2009, *Nature chemical biology* 5, 593; Park, J.O., Rubin, S.A., Xu, Y.F., Amador-Noguez, D., Fan, J., Shlomi, T. and Rabinowitz, J.D., 2016. *Nature chemical biology*, 12, 482.) found that in *E. coli* metabolite concentration exceeds K_i for most substrate-enzyme pairs. i.e. $[S_i] > K_i$,

which suggests that we can use k_i to roughly estimate the value of κ_i . Furthermore, in the simplest case with no complex regulations (SI Sec. 1.4) or in more complexed situations such as reversible reactions, cooperative effect, and reversible product inhibition (SI Sec. 6.1-6.3) that are common in metabolic network, $\kappa_i = k_i$ is the optimal solution. Since biochemical parameters of enzyme kinetics considering metabolic regulations cannot be collected (mostly have not been measured) from literatures at the present stage, $\kappa_i \approx k_i$ can be a rough estimation for the value of κ_i , especially for the purposes of assigning carbon sources to the precursor pools as it is determined by the topology of the metabolic network more than the precise parameter values. Our experimental results of the carbon supply pattern indicated that the carbon sources can be grouped into subgroups and the same subgroup show the same carbon percentage in precursor pools. This strongly suggests that the requirement on precise parameter values can be relaxed.

2) Starting from line 270 in the SI, the authors state that there is currently no experimental data to compare with the predictions in Supplementary Table 3, which is a prediction of which resource pool is supplied by which carbon source. Since this is the most predictive statement of the whole work, this demands an experiment. It could for example be tested if the reactions connecting two of the pools are actually switched off when growing on multiple substrates.

Response: We thank the reviewer for the suggestion of doing experiments. We quantitatively measured the percentages of carbon in representative amino acids coming from each carbon source, using isotope labeling experiments. The coupling between theory and experiments prompted us to further improve the model (mainly taking into account the TCA cycle in carbon flow (see SI Sec. 4 for details)). Our model predictions now quantitatively agree with all experimental results (Figs. 3-4, S5-S6 and Table S4).

3) The authors dedicate a section in the main text to data from Escalante-Chong et al. This data shows heterogeneity in a yeast population that switches from galactose to glucose. Escalante-Chong et al. have already concluded that *S. cerevisiae* decides which carbon source to consume depending on the ratio between the sources. They find an almost straight 'switching line' when galactose (%) is plotted versus glucose (%). That the authors can fit this straight line using 5 parameters can hardly be called 'remarkable'. The remarkable aspect of this experimental dataset is the heterogeneity in the population. If the authors could explain this from an optimization perspective, then it would add something valuable to the existing theory of biological optimization.

Response: We thank the reviewer's criticism and suggestion. We have moved this part to SI (SI Sec. 2.3). Actually, there are 2 rather than 5 parameters (δ and Δ in Eq. S14) to fit the turning points in Fig. S2 ($R^2=0.995$), the slope=1 (when both concentrations are low) is a parameter free prediction.

Our theory explains the overall switching line (50% switching in a population) in Fig. S2. Explaining heterogeneity itself is a very interesting problem and is out of the scope of this study.

Reviewer #2 (Remarks to the Author):

When grown on a medium containing two carbon substrates, microbes often take up just one, resulting in the famous phenomenon of diauxie. Based on the fact that the elected substrate is usually the one that provides the faster growth, many authors have speculated that this behavior is in fact optimal: assuming that all metabolic precursors required for growth can be derived from either carbon substrate, it makes sense to just consume the substrate that does so with the smallest investment. Previously, authors typically assumed that the "currency" of the investment was "energy"; in the light of the recent work of the Hwa group and others, it makes sense to replace this with "proteome fraction", but the argument stays the same.

While this old argument sounds plausible, it does not explain the basic fact that, in *E. coli* and other microbes, many substrate pairs are in fact co-utilized. Why doesn't the argument hold for these cases?

This manuscript goes beyond the old hypothesis by taking into account that the efficiency of each carbon substrate (in terms of investment of proteome) actually differs for each molecular precursor (e.g., different amino acids) required for growth. They show that, by taking this into account, it becomes clear that the optimal solution (the one maximizing the growth rate) then depends on the entry point of each substrate into the metabolic network, and on the topology of the network itself.

Many good ideas are obvious in retrospect. This is such an idea. The authors have convinced me that this argument indeed an important factor in the distinction between co-utilized and sequentially utilized substrate pairs.

An important aspect in favor of the theory is that it makes predictions that are rather easy to verify experimentally, and the authors do in fact provide such predictions (shown in Supp. Tables). It is a bit of a shame that the authors did not team up with experimentalists in order to verify these predictions, but I for one believe that publishing theoretical predictions before their experimental verification should become more common in biology -- this practice has served other disciplines (physics, chemistry) well for the past few centuries.

Response: We thank the reviewer for the very positive comments. We have followed many of the suggestive comments raised by the reviewer and the manuscript is much improved as a result.

Most importantly, following the reviewer's suggestion, we teamed up with experimentalists to test the model. We quantitatively measured the percentages of carbon in representative amino acids coming from each carbon source, using isotope labeling experiments. The coupling between theory and experiments also prompted us to further improve the model (mainly taking into account the TCA cycle in carbon flow (see SI Sec. 4 for details)). Our model predictions now quantitatively agree with all experimental results (Figs. 3-4, S5-S6 and Table S4).

Some points that the authors should (be able to) address:

* In figure 1, glycerol is categorized into group B. It is known, however, that glycerol is not co-utilized with glucose or lactose. Can the model explain this? If the authors "predict" the utilization pattern for glucose + glycerol with their model (as in Supplementary Table 3), what do they find?

Response: We thank the reviewer for this suggestive comment. In the case in which glycerol is mixed with glucose or lactose all in saturated concentrations, based on the branch efficiencies calculated in Table S2, glucose or lactose owns higher branch efficiencies than glycerol for all precursor pools, so our model predicts *E. coli* will only use glucose or lactose, which is consistent with experiment.

On the other hand, when the concentration of glucose or lactose is low, based on the network topology, glycerol can be co-utilized with glucose or lactose, and actually, this was recently observed in experiment (Okano H, Hermsen R, Kochanowski K, Sauer U, Hwa T. Regulation of hierarchical and simultaneous carbon-substrate utilization by flux sensors in *Esherichia coli*. *Unpublished*). To avoid confusion, we now classify glycerol as a quasi-Group A carbon source, and we have added a note in the caption of Fig. 1.

* The results in Fig. 5 show some correlation between the predictions and the actual uptake measurements, but I'm not too impressed by the predictions for mixture 2 and 3. Why and how do the authors decide that the predictions are "consistent" with the experimental results?

Response: In our original manuscript, we did not have experimental data to directly compare with model predictions in Table S4, so we used this indirect comparison in the old Fig. 5. The discrepancies between the predictions and the uptake measurements were due to the fact that we did not consider the influence of energy production in our old version (TCA cycle). Now, by quantitatively estimating the effect of energy production, we improved our theory (see SI Sec.4 for details). Since we now have our own experimental data for direct comparison with the model predictions (Table S4), we have removed the old Fig. 5.

* Generally speaking, the manuscript is clearly written and the figures/illustrations have been carefully drawn. The Discussion section, however, should definitely be improved. Currently, it reads as an incoherent collection of statements and topics that had to be mentioned somewhere. For instance, what is the main message of the first paragraph? And what is the point of the paragraph starting with "Cell growth is a fundamental issue in biology."? What does that sentence even mean?

Response: We thank the reviewer for the comments. We have reorganized our discussion section. In the first paragraph, we discuss results and relevant studies involving diauxie; in the second paragraph, we discuss results about co-utilization; the last two paragraphs are dedicated to discussions on the applicability, limitation and exceptional cases of our theory.

* Here and there, I found the mathematical notation to be unnecessarily complex. In line 83, for instance, the indices within indices can easily be avoided, and the protein mass of the cell could simply be called M. Throughout, indices "Ai" could, I believe, simply be replaced with "i". The point is: I field that the authors should try to minimize and simplify the mathematical expression in the main text, obviously without compromising the main message, in order to ensure that the paper is accessible to the wide readership of this journal.

Response: We thank the reviewer for the comments. We rewrote most part of our Main-Text and Supplemental Information. Now our manuscript should be more accessible to the wide readership. We still kept indices 'A1' and 'A2', since in such cases both carbon sources are from Group A.

Reviewer #3 (Remarks to the Author):

In this manuscript, the authors aim to understand the two different carbon utilizing behaviors exhibited by bacteria growing on carbon mixtures: sequential utilization (diauxie) and simultaneous utilization (co-utilization). With mathematical analysis of coarse-grained metabolic models, they demonstrated that the two different behaviors originate from the principle of optimal protein resources allocation for maximum growth. The study excels at abstracting simple coarse-grained models (e.g., Fig. 2c and Fig. 4b) from the complicated metabolic network. These concise but biologically relevant models allow rigorous mathematical analysis for biological insights. This work is an example of first-class theoretical biology studies. There are however a few points that need to be clarified before being published, particularly about the comparisons between model predictions and experimental data. I also have some suggestions for improving the manuscript.

Response: We thank the reviewer for the very positive comments. We also thank the reviewer for carefully reading the manuscript and suggestions to improve the paper.

Major:

1. The authors mentioned a comparison between the theory and experiment with *Methylobacterium extorquens* AM1 (Page 13). I however couldn't locate such a comparison. It is not in the SI.

Response: We apologize that we did not make it clear enough. Now we have specified in the SI Sec. 4.9 that the experimental data with *Methylobacterium extorquens* AM1 is listed in Figure 3 of the reference article (Peyraud R, Kiefer P, Christen P, Portais J-C, Vorholt JA. 2012, *PLoS One* 7, e48271). We emphasize that this is a qualitative comparison, similar with the pattern shown in our Fig. S4a. We now have added quantitative experiments to compare with the model predictions (see Figs. 3-4, S5-S6 and Table S4).

2. It is not clear what is exactly being compared in Fig. 5. How were the "experimental results" obtained? These numbers are not directly available in the reference 6. Were they calculated using Eqn. S43? But according to the text after Eqn. S43, the results from the equation are not meant to be "experimental results" because "they are plotted in Fig. 5 along with the experimental data" (Line 308 in the SI). What about the "model predictions"? Were they calculated utilizing the numbers in Line 281 and Line 282 in the SI? In sum, it seems that the main text and SI are contradictory about Fig. 5; and in any case the text about Fig. 5 is too brief.

Response: We apologize that we did not make it clear in our original version of the manuscript. Experimental results in Fig. 5 (in our old version) were calculated using Eqn. S43 (old version), while λ_A , λ_B , η_A , η_B , η'_A and η'_B in Eqn. S43 (old version) were experimental data obtained from Hermsen R, *et.al*, *Molecular systems biology* 11, 801. Model predictions in

Fig. 5 (old version) were exactly calculated using the numbers in Line 281 and Line 282 in the SI (old version).

We quantitatively measured the carbon supply percentage of amino acids with isotope labeling experiments, which corresponds to the model predictions in Table S4. Happily, our model predictions quantitatively agree with all experimental results (see Figs.3-4, S5-S6 and Table S4). Now that we have our own experiments for direct comparison, we removed old Fig. 5 which was an indirect comparison.

3. In Eqns. 1 and 2, more explanations on the inequalities are needed. For example, doesn't flux balance always enforce equality in Eqn. 2? Is the inequality due to flux leakage? And the inequality in Eqn. 1 depends on how ϕ_{\max} is defined.

Response: Yes, flux balance always enforces equality, and the inequality is due to potential leaks which were not considered in our study. To avoid this confusion, we have delete ϕ related denotations in our manuscript and rewrote most part of our Main-Text and SI.

Minor:

4. While referring to the SI, instead of saying the same "see SI for details" through the text, the authors should provide more specific locations, e.g., see Sec. 5 of SI for details.

Response: We thank the reviewer for the suggestion, and we now have provided specific locations in the main-text and SI.

5. I find the use of "nutrient quality", "nutrient efficiency", and "substrate quality" confusing. They have the same units. Is it possible to use the same term for all of them? Along the same line, is it possible to use a single symbol for nutrient efficiency (ϵ) and nutrient quality (κ)?

Response: We thank the reviewer for the suggestion. We stopped using "nutrient quality" and "nutrient efficiency", and only use "substrate quality". Efficiency ϵ and substrate quality κ share the same units. However, κ is used to denote the efficiency of the specific enzyme reaction with that substrate, while ϵ is used to denote the overall efficiency of a substrate. Using a single symbol may cause misunderstandings when we compare the overall efficiency of two substrates (e.g. κ_{A1} can be smaller than κ_{A2} when ϵ_{A1} is large than ϵ_{A2} in model shown in Fig. S1c).

6. In Line 87, k_{Ai} and K_{Ai} are not defined.

7. In Line 129, k_M should be κ_M .

Response: We have fixed these issues and have rewritten our manuscript and SI.

8. In Line 145, instead of using the subtly different k_{A1}^{A2} and K_{A1}^{A2} , consider use a

different symbol other than k for them.

Response: Now we use δ (delta) to replace k_{A1}^{A2} and Δ (Delta) to replace K_{A1}^{A2} .

9. In Line 34 of SI, what is the physical/biological meaning of the coefficient alpha?

Response: We have rewritten our manuscript and SI and stopped using alpha.

10. In Line 139 of SI, ϕ_R should be ϕ_S .

11. In Line 172 of SI, k_M should be κ_M .

12. In Line 178 of SI, k_M should be κ_M .

13. In Line 189 of SI, it is a jump from κ_{Ai}^j in Eqn. S30 to k_{Ai}^j in Eqn. S32. The approximation should be explicitly indicated during the derivation.

Response: We have fixed these issues and rewrote our manuscript and SI.

Reviewers' comments:

Reviewer #2 (Remarks to the Author):

In response to the points raised by the reviewers, the authors have substantially extended their theoretical analysis and have performed experimental tests to validate their predictions. As a result, both the main text and the supplement have been rewritten to a large extent.

With this extra effort, I believe the authors have adequately addressed my own comments:

- I commented that the authors could have teamed up with experimentalists to verify their predictions. They now include measurements of the percentages of the carbon in various amino acids that come from different carbon sources (using isotope labeling experiments). These experiments are a valuable addition to the paper.
- My question about glycerol uptake was adequately answered.
- Fig. 5 has now been replaced with more direct data.
- The discussion section has been rewritten, as I suggested.
- The mathematical notation in the paper and the supplement has improved.

I have been asked to also comment on the way the authors dealt with the comments by reviewer 1.

- The main criticism of reviewer 1 was that the theory presented here is a special case of an existing more general theory (the theory on EFMs) that does not require some of the simplifying assumptions that the authors make. In the previous version of the manuscript, this theory was not even mentioned, which was inappropriate, in particular because the authors did know about this work.

That said, the authors now argue that the theory on EFMs (in particular the theorem that the optimal flux distribution is an EFM) is insufficient to reproduce their results. For a broad class of models, the existence of an EFM in which both carbon sources are utilized is a necessary condition for the co-utilization of these carbon sources; but it is not a sufficient condition if other EFMs exist in which just one carbon substrate is used, because in that case it remains to be determined which EFM is optimal – which depends on the costs of each reaction.

I believe that, on balance, I agree with reviewer 1 that the EFM papers had to be cited and discussed, which the authors now do. But I agree with the authors that (a) purely EFM-based theory is not entirely sufficient to resolve the puzzle of co-utilization vs. sequential co-utilization, and (b) in as far as it is sufficient, none of the EFM papers (to my knowledge!) applies the theory to this puzzle.

- Specific comment 1:

The reviewer was correct that the statement on the κ_M in the previous version of the manuscript was incorrect. The authors have performed quite some extra analysis to fix this. They still make some rather strong assumptions about the saturation of enzymes, and I'm not sure reviewer 1 would be completely satisfied. My own opinion is, however, that these simplifying assumptions (in particular that most enzymes are approximately saturated) serve to communicate a general point (that, due to the topology of the network, certain combinations of carbon sources can be co-utilized under optimality, while other combinations cannot) that is likely to hold up for more complex models that do not make these assumptions. As such, I care less about these assumptions at this point, as long as they are clearly stated.

- Specific comment 2:

The authors have now included such experiments.

- Specific comment 3:

The reviewer's remark that fitting a straight line with multiple parameters is not a great feat is well taken, even though the authors' response makes it clear that the fitting is a little more constrained than the reviewer makes it sound. I do not think, however, that it is reasonable to ask the authors to make predictions about population heterogeneity, which requires an entirely different class of models and would not contribute much to the central message of the manuscript.

Reviewer #3 (Remarks to the Author):

With extensive new experimental data, the revised manuscript is a further improvement. Given that the direct comparison between experiment and theory is now a substantial part of the paper, I have a question regarding the enzymatic parameters used for model predictions. The k_{cat} value of an enzyme depends on the exact environment (e.g., temperature, pH, etc) where the reaction takes place. Thus the measured k_{cat} values are perhaps associated with large errors. Can the authors estimate errors for the k_{cat} values in Table S1? The errors may have strong influences on the predictions as the substrate branch efficiencies are (surprisingly to me) not very different (<2-fold) from each other in most cases (Table S2). In other words, the authors should put error bars on the empty bars in Fig. 3 and Fig. 4.

In Line 108 of the main text, "qualify" should be "quality".

In Line 161 and Line 162 of the SI, the parameters " K_{A1}^{A2} " and " k_{A1}^{A2} " did not occur in the equations.

In Line 381 and Line 382 of the SI, please fix the grammatical problems in the sentence.

Response to reviewers

(Reviewers' comments in black and our response in blue)

Reviewers' comments:

Reviewer #2 (Remarks to the Author):

In response to the points raised by the reviewers, the authors have substantially extended their theoretical analysis and have performed experimental tests to validate their predictions. As a result, both the main text and the supplement have been rewritten to a large extent.

With this extra effort, I believe the authors have adequately addressed my own comments:

- I commented that the authors could have teamed up with experimentalists to verify their predictions. They now include measurements of the percentages of the carbon in various amino acids that come from different carbon sources (using isotope labeling experiments). These experiments are a valuable addition to the paper.
- My question about glycerol uptake was adequately answered.
- Fig. 5 has now been replaced with more direct data.
- The discussion section has been rewritten, as I suggested.
- The mathematical notation in the paper and the supplement has improved.

I have been asked to also comment on the way the authors dealt with the comments by reviewer 1.

- The main criticism of reviewer 1 was that the theory presented here is a special case of an existing more general theory (the theory on EFMs) that does not require some of the simplifying assumptions that the authors make. In the previous version of the manuscript, this theory was not even mentioned, which was inappropriate, in particular because the authors did know about this work.

That said, the authors now argue that the theory on EFMs (in particular the theorem that the optimal flux distribution is an EFM) is insufficient to reproduce their results. For a broad class of models, the existence of an EFM in which both carbon sources are utilized is a necessary condition for the co-utilization of these carbon sources; but it is not a sufficient condition if other EFMs exist in which just one carbon substrate is used, because in that case it remains to be determined which EFM is optimal – which depends on the costs of each reaction.

I believe that, on balance, I agree with reviewer 1 that the EFM papers had to be cited and discussed, which the authors now do. But I agree with the authors that (a) purely EFM-based theory is not entirely sufficient to resolve the puzzle of co-utilization vs. sequential co-utilization, and (b) in as far as it is sufficient, none of the EFM papers (to my knowledge!) applies the theory to this puzzle.

- Specific comment 1:

The reviewer was correct that the statement on the κ_M in the previous version of the manuscript was incorrect. The authors have performed quite some extra analysis to fix this. They still make some rather strong assumptions about the saturation of enzymes, and I'm

not sure reviewer 1 would be completely satisfied. My own opinion is, however, that these simplifying assumptions (in particular that most enzymes are approximately saturated) serve to communicate a general point (that, due to the topology of the network, certain combinations of carbon sources can be co-utilized under optimality, while other combinations cannot) that is likely to hold up for more complex models that do not make these assumptions. As such, I care less about these assumptions at this point, as long as they are clearly stated.

- Specific comment 2:

The authors have now included such experiments.

- Specific comment 3:

The reviewer's remark that fitting a straight line with multiple parameters is not a great feat is well taken, even though the authors' response makes it clear that the fitting is a little more constrained than the reviewer makes it sound. I do not think, however, that it is reasonable to ask the authors to make predictions about population heterogeneity, which requires an entirely different class of models and would not contribute much to the central message of the manuscript.

Response: We thank the reviewer for the very positive comments. We also thank the reviewer for efforts and the very positive comments on the way we dealt with the comments by reviewer 1.

Reviewer #3 (Remarks to the Author):

With extensive new experimental data, the revised manuscript is a further improvement.

Response: We thank the reviewer for the very positive comments.

Given that the direct comparison between experiment and theory is now a substantial part of the paper, I have a question regarding the enzymatic parameters used for model predictions. The k_{cat} value of an enzyme depends on the exact environment (e.g., temperature, pH, etc) where the reaction takes place. Thus the measured k_{cat} values are perhaps associated with large errors. Can the authors estimate errors for the k_{cat} values in Table S1? The errors may have strong influences on the predictions as the substrate branch efficiencies are (surprisingly to me) not very different (<2-fold) from each other in most cases (Table S2). In other words, the authors should put error bars on the empty bars in Fig. 3 and Fig. 4.

Response: We thank the reviewer for raising this question. In the first revision, we did briefly mention that the model prediction is somewhat robust to parameters due to "quantization" of carbon source utilization patterns. We now realize that a more detailed analysis is needed to substantiate this point. We have greatly extended Supplementary Sec. 4.9 to address this issue, which we summarize below.

While it is difficult to estimate the errors of \$k_{cat}\$ taken from the literature, one can look at

this problem from an opposite point of view: consider all possible values of k_{cat} . For example, consider the case of A_1+B_1 , where A_1 is a group A source and B_1 a group B source. Taking all possible values of k_{cat} (so long as $k_{cat}>0$) for all enzymes involved, we obtain N possible utilization patterns (N ranges from 10-20; see Supplementary Table 7). Among these patterns, only a few are qualitatively similar to the experimentally observed one, say Patterns X, Y and Z, and one of which (say Z) *quantitatively* agrees the experimental result. The nominal parameters of k_{cat} taken from the literature would give the solution Z. The parameters that would give rise to solution X or Y would be significantly different (5-30 folds change) from the literature values. Note that the above analysis works consistently for all combinations of $A+B$ (and a case of $B+B$), with the same set of k_{cat} 's from the literature. There are only a couple of exceptions: in the cases of k_{cat} for some transporters (for which there is no experimental value), a reduction of 40% can switch the pattern (e.g. from Z to X). However, such tuning can hardly fit the patterns for all combinations.

We have added a few sentences and referred this analysis in the Discussion section of the main text to highlight the robustness of model prediction.

In Line 108 of the main text, "qualify" should be "quality".

In Line 161 and Line 162 of the SI, the parameters " $K_{A_1A_2}$ " and " $k_{A_1A_2}$ " did not occur in the equations.

In Line 381 and Line 382 of the SI, please fix the grammatical problems in the sentence.

Response: We thank the reviewer for carefully reading the manuscript and we have fixed these issues.

REVIEWERS' COMMENTS:

Reviewer #3 (Remarks to the Author):

The authors have satisfactorily addressed all my concerns.

Response to reviewers

(Reviewers' comments in black and our response in blue)

Reviewer #3 (Remarks to the Author):

The authors have satisfactorily addressed all my concerns.

Response: We thank the reviewer for the very positive comments.